# Trend of influenza before and during the COVID-19 pandemic in Nepal—A study from 2018 to 2022

Smriti Shrestha[1][☉], Priya Jha[2][☉]*, Lilee Shrestha[1], Lok Bandhu Chaudhary[1], Rashmi Mulmi[2], Arunkumar Govindakarnavar[2], Runa Jha[1]

**1** Department of National Influenza Center, National Public Health Laboratory, Kathmandu, Nepal, **2** WHO Health Emergencies Programme, World Health Organization Country Office for Nepal, Lalitpur, Nepal

☉ These authors contributed equally to this work.
* pjha@who.int

## Abstract

A significant reduction in influenza incidence during the early days of COVID-19 pandemic was reported worldwide. This study aims to understand the impact of public health and social measures implemented during the COVID- 19 pandemic on influenza circulation in Nepal. We utilized influenza sentinel and non-sentinel surveillance data from Nepal between 2018 and 2022, obtained from the National Influenza Centre (NIC) at National Public Health Laboratory (NPHL), Nepal. Additionally, we used publicly available national COVID-19 case data released by the Ministry of Health and Population of Nepal. The data were analyzed for the trends in influenza cases, positivity rate and the distribution of subtypes/lineages. Furthermore, we compared the trend of influenza with that of COVID-19 and the social and public health measures implemented in the country as part of the COVID-19 response. The average influenza positivity rate dropped significantly from 39% to 14% during the COVID-19 period compared to the pre- COVID-19. Additionally, during the time of COVID 19 there has been a shift in the influenza bimodal seasonal pattern, with only one peak observed. Influenza type A consistently dominated, with variations in its subtype observed from year to year. Notably, one case of Influenza A/H5N1 was reported in 2019. This study shows that the influenza positivity rate decreased substantially after the COVID-19 pandemic began, possibly due to the stringent public health and social measures implemented during the pandemic. Adaptation of the influenza surveillance system during pandemics and integration of other respiratory pathogens into it not only allows continuity of surveillance but also helps to evaluate the public health and social measures implemented to manage future respiratory virus pandemics.

**Data availability statement:** The URL of the RespiMart output, from which the data for this research was extracted, has been shared and can be accessed at https://www.who.int/teams/global-influenza-programme/surveillance-and-monitoring/influenza-surveillance-outputs. Additionally, anonymized data with demographic information is provided as Supporting Information files to replicate the figures and tables.

**Funding:** The author(s) received no specific funding for this work.

**Competing interests:** No.

## Introduction

The influenza virus is well known for the pandemics it has caused and its seasonal activity [1]. The World Health Organization (WHO) estimates that the influenza virus infection alone results in between 290,000 and 650,000 fatalities annually, with 36% of these deaths taking place in low- and middle-income nations [2]. Furthermore, it is estimated that around 20% of the world's population is exposed to influenza virus each year, with 20–30% in pediatric and 5–10% among adult populations [3]. In Southeast Asia, the average annual death rate associated with influenza is notably high, ranging from 3.5 to 9.2 per 100,000 people [4].

Coronavirus disease 2019 (COVID-19), caused by severe acute respiratory syndrome coronavirus (SARS-CoV-2), was first reported in Wuhan, China, in December 2019 [5]. The World Health Organization (WHO) declared the outbreak of COVID-19 Public Health Emergency of International Concern (PHEIC) on January 30, 2020, and a pandemic on March 11, 2020 [6]. In Nepal, the first case of COVID-19 was confirmed in January 2020, followed by a second case in March 2020, with a sudden increase observed in May 2020 [7]. The first wave of COVID-19 in Nepal peaked in late October 2020, the second wave due to delta variant in May 2021, and the third wave peaked in January 2022 [8,9]. The Public Health and Social Measures (PHSM) such as lockdown, and travel restrictions, severely affected people's mobility in Nepal 2020 and the first half of 2021 [10]. These measures, implemented to control and mitigate the COVID-19 pandemic, significantly impacted influenza surveillance globally, including in Nepal. Many countries that maintained influenza surveillance reported a decline in seasonal influenza activity during early days of the COVID-19 pandemic [11]. While sentinel influenza surveillance activities in Nepal also significantly reduced during early days of COVID-19 pandemic, it was supplemented through testing SARS-CoV-2 PCR negative samples for influenza following the WHO guideline [12].

The National Influenza Center (NIC) at the NPHL has been conducting influenza surveillance at the national level since 2010. This is a hospital-based sentinel surveillance program including 15 hospitals selected based on geographical region and population density with a provision for sending representative samples every week. Similarly, other collaborating laboratories (non -sentinel) share their aggregate influenza test result with NIC for upload to the RespiMart platform. RespiMart, developed by FluMart is a data sharing platform where countries participating in the Global Influenza Surveillance and Response System (GISRS) share both aggregated and case-specific data related to respiratory viruses capable of causing epidemics and pandemics. These includes influenza, respiratory syncytial virus (RSV) and SARS-CoV-2 virus. It facilitates the exchange and storage of surveillance data enabling integrated analysis and reporting [13]. The National Influenza Center (NIC) is responsible for representing Nepal in RespiMart, maintaining a database of influenza tests performed at NPHL and other collaborating centers in MS-Excel, and uploading the data to RespiMart weekly.

We utilized this platform to analyze the influenza surveillance data from Nepal for the period 2018–2022 to understand influenza trends before and during the

COVID-19 pandemic. This study aims to understand the impact of public health and social measures implemented during the COVID-19 pandemic on influenza circulation in Nepal from 2018 to 2022.

## Materials and methods

This retrospective cross-sectional study was conducted using influenza data from Nepal reported to RespiMart between 2018 and 2022. The period from 2018 to 2019 was considered the pre-COVID-19 period, while from 2020 to 2022 was considered the COVID-19 period. A total 26,603 samples were collected from 15 influenza sentinel and non-sentinel sites across the eastern, western, central and far-western regions of Nepal. These sites were selected based on regional representation, geographical conditions, population density and areas prone to potential epidemics to ensure the provision of representative influenza samples to the NIC. The data were analyzed using MS Excel to determine the total number of samples processed, the influenza positivity rates, and the distribution of influenza types (influenza A and B), and subtypes (influenza A(H1N1) pdm09 and A(H3N2), and B lineages, (Victoria and Yamagata) across epidemiological weeks. Monthly cases distributions were also calculated to understand the seasonal influenza pattern before and during the COVID-19 period. The Chi-square test was applied to test the association between influenza positivity in pre and during COVID-19 period. The COVID-19 trend was plotted and compared with the influenza trend. To analyze the distribution of influenza across various age groups, genders, and case types—Severe Acute Respiratory Infection (SARI) and Influenza-like Illness (ILI) we used the electronic data of specimens tested at the NIC only, as epidemiological data from collaborating centers were not received.

Case Definition of SARI and ILI as per WHO case definition:

Severe Acute Respiratory Infection (SARI): Cases with an acute respiratory infection with history of fever or measured fever of ≥ 38°C and cough; with onset within the last ten days and requires hospitalization is defined as SARI.

Influenza-like Illness (ILI): Cases with an acute respiratory infection with history of fever or measured fever of ≥ 38°C and cough; with onset within the last ten days is defined as ILI.

For the detection and quantification of influenza viruses in clinical samples, a highly sensitive and specific molecular diagnostic technique, Real-time polymerase chain reaction (RT-PCR) was employed. Respiratory specimens, such as nasal swabs or throat swabs were collected from symptomatic individuals suspected of having flu-like symptoms. These samples were then processed in a molecular laboratory setting where viral RNA was extracted and purified using the QIAmp Viral RNA Mini Kit (QIAGEN, Valencia,CA,USA). The extracted nucleic acid was then subjected to PCR amplification for detection of influenza A and influenza B using influenza virus Real- Time A/B typing Kit (Center For Disease Control and Prevention, Atlanta, GA, USA). Further subtyping of positive influenza A samples were performed by using CDC influenza Virus Real-Time RT-PCR panel, influenza A (H3N2/H1N1pdm09) subtyping panel (CDC and the influenza B Lineage Genotyping panel (CDC) to detects the lineage of the influenza B virus).

Ethical approval was obtained from Nepal Health Research Council (Approval ID: 572–2022) to conduct the study. The data were accessed and analyzed in 2023 (March to May)

## Result

A total of 26,603 samples were reported in RespiMart from 2018 to 2022. However, age and gender data, clinical history were available only from 18688 cases (Table 3). The average influenza positivity rate was 39% pre- COVID-19, compared to 14% during COVID-19, indicating a significant reduction (p value < 0.001at 95% CI) once the COVID-19 pandemic began in Nepal as shown in Table 1.

## Distribution of influenza type and subtype

Fig 1 illustrates the trend in the prevalence and distribution of influenza across different years, comparing the pre-COVID-19 period (2018–2019) with the period during COVID-19 (2020–2022). Table 2 demonstrate, Influenza A was more

**Table 1. Total number of samples tested and positivity rate of Influenza circulation in pre-COVID-19 and during COVID-19.**

|  | Year | Total Sample (N) | Influenza Negative N (%) | Influenza Positive N (%) | (%) Influenza Positivity N (%) | P-Value |
|---|---|---|---|---|---|---|
| Pre-COVID-19 | 2018 | 3091 | 2134(69) | 957(31) | 3526 (39.0) | <0.001 |
|  | 2019 | 5783 | 3214(55.5) | 2569(44.4) |  |  |
| DURING COVID-19 | 2020 | 2915 | 2223(76.2) | 692(23.7) | 2469 (14.0) |  |
|  | 2021 | 6777 | 5642(83.2) | 1135(16.7) |  |  |
|  | 2022 | 8037 | 7332(91.2) | 705(8.7) |  |  |

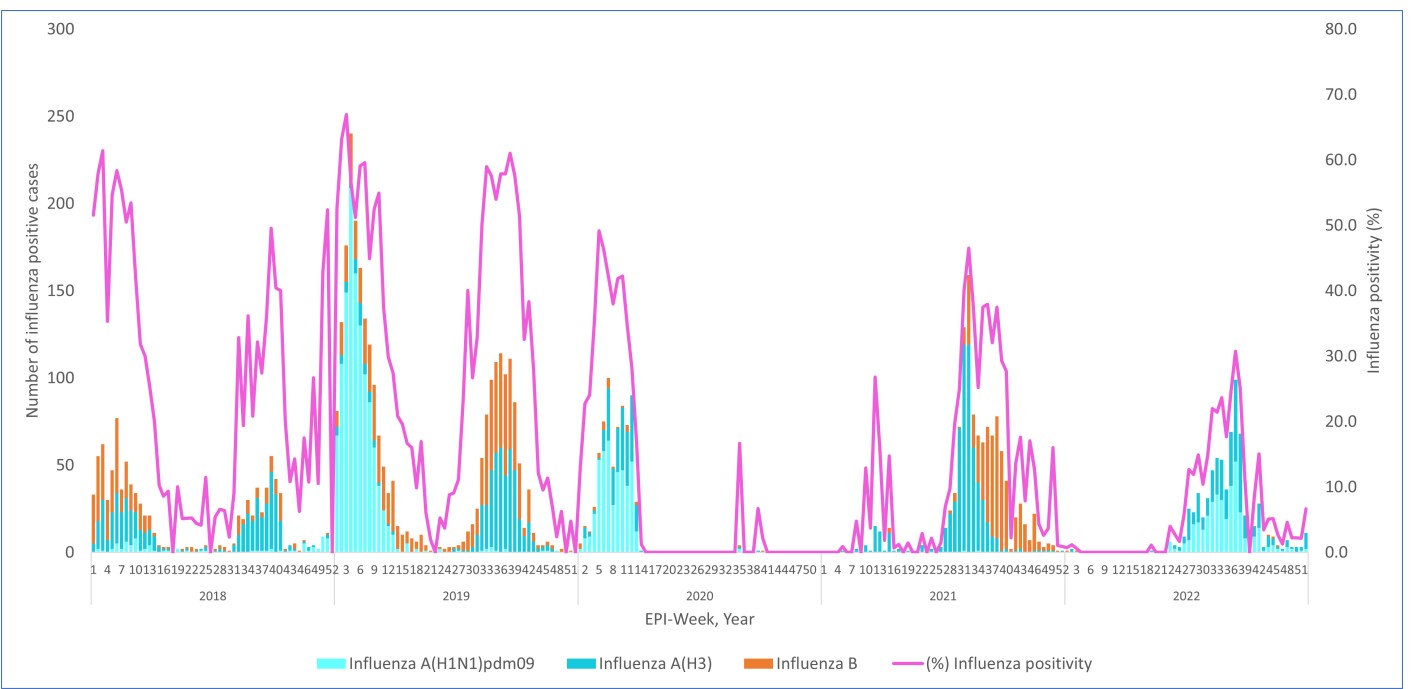

**Fig 1. Trend of Influenza during Pre-COVID-19 (2018-2019) and during COVID -19 (2020–2022).**

prevalent than influenza B throughout the given period. During the pre-COVID-19 period, 26% (2274/8874) of influenza cases were type A, compared to 14% (1252/8874) of influenza type B cases among the total tested samples. During the COVID-19 period, 11.1% (1,971/17,729) of cases were influenza A, and 3.16% (561/17,729) were influenza B, indicating a significant drop in the overall influenza positivity rate. A significant reduction (P < 0.001) in influenza positivity was observed from the pre-COVID-19 to the COVID-19 period.

As illustrated in Fig 2, among the subtypes of influenza A, Influenza A(H3N2) was dominant strain in 2018 and 2021, while influenza A(H1N1) Pdm09was dominant in 2019 and 2020, Influenza A(H1N1)Pdm09 and influenza A(H3N2) were codominant in 2022. Remarkably, one influenza A(H5N1) subtype was reported in 2019.

## Monthly pattern of influenza positivity from 2018 to 2022

As shown in Figs 3 and 4, a distinct pattern of peaks was observed during two study periods: pre-COVID-19 and during COVID -19. In the pre-COVID -19 years, influenza positivity was observed throughout the year, with two peaks- one from

**Table 2. Influenza Types before and during the COVID-19 pandemic.**

| | Year | Influenza Type A Positive* | H1N1pdm09 | H3N2 | H5N1 | Not Sub-typed | Influenza Type B Positive* | Victoria Lineage | Yamagata Lineage | Lineage not determined |
|---|---|---|---|---|---|---|---|---|---|---|
| | | N (%) | N (%) | N (%) | N (% | N (%) | N (%) | N (%) | N (%) | N (%) |
| Pre-COVID-19 | 2018 | 574(59.97) | 83(8.9) | 491(52.7) | 0 | 0 | 383(40.02) | 13(1.3) | 175(18.8) | 195(20.9) |
| | 2019 | 1700(66.17) | 1183(46) | 498(19.3) | 1(0.04) | 18(0.7) | 869(15.02) | 63(2.4) | 24(0.9) | 782(30.4) |
| During COVID-19 | 2020 | 661(95.52) | 438(63.2) | 223(32.2) | 0 | 0 | 31(1.06) | 0 | 0 | 31(4.47) |
| | 2021 | 610(53.7) | 2(0.17) | 608(53.5) | 0 | 0 | 525(7.74) | 235(20.7) | 0 | 290(25.5) |
| | 2022 | 700(99.2) | 362(51.3) | 338(47.9) | 0 | 0 | 5(0.06) | 1(0.14) | 0 | 4(0.56) |

*For calculating the % of influenza types and subtypes, the denominator used is the total number of influenza positive cases.

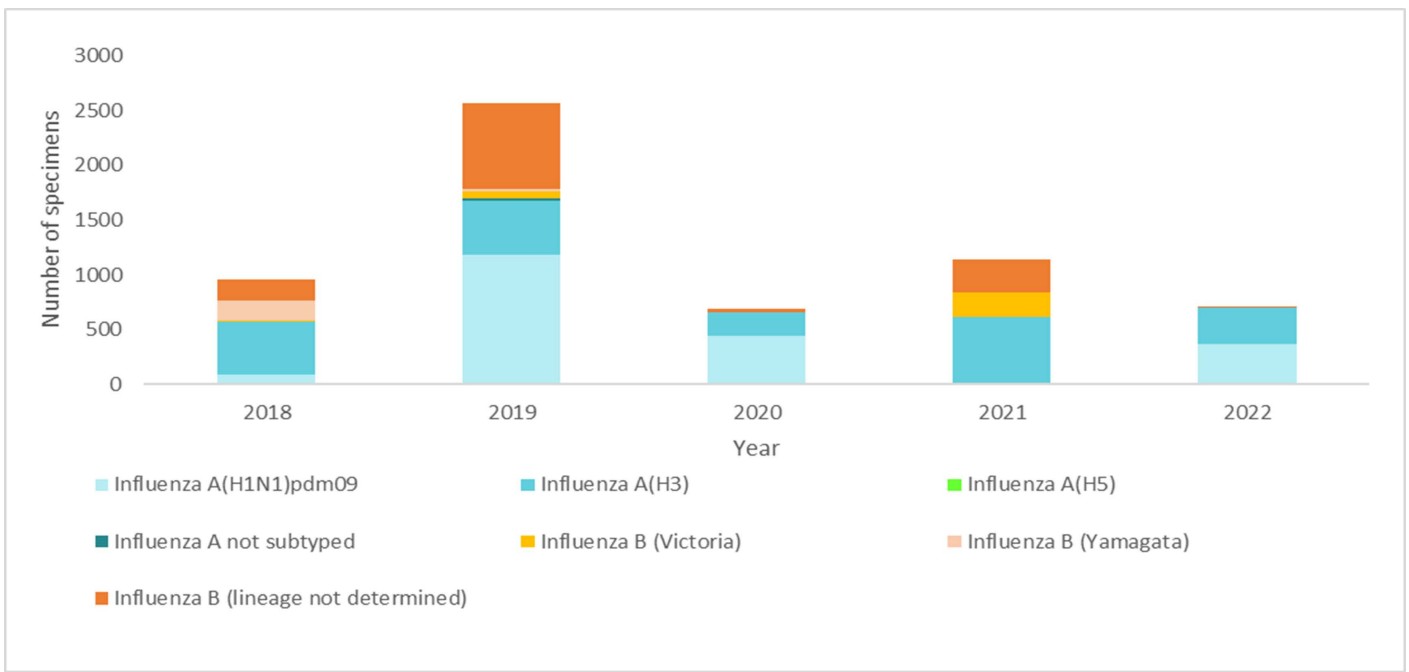

**Fig 2. Circulation of influenza A subtype and influenza B Lineage from 2018 to 2022.**

January to March and another from July to October. In contrast, during the COVID-19 years (2021 and 2022), only a single peak was observed from July to October. In 2020, influenza positivity was observed only from January to March. As shown in Fig 5, COVID-19 cases in Nepal began to rise from epidemiological week 21 in 2020, during which no influenza cases where reported. Similarly, in 2021 and 2022, a decrease in influenza incidence coincided with an increase in COVID-19 cases, while an increase in influenza cases coincided with a decrease in COVID-19 incidence.

### Age, gender and case wise distribution of influenza

As described in Table 3, the influenza positivity rate was mainly observed in the age group (15–64) during pre-COVID-19, while it was higher in pediatric age group<5 followed by 5–14 years during the pandemic period. The case-wise distribution in the study shows that ILI cases were higher compared to SARI cases during pre-COVID-19 period. However,

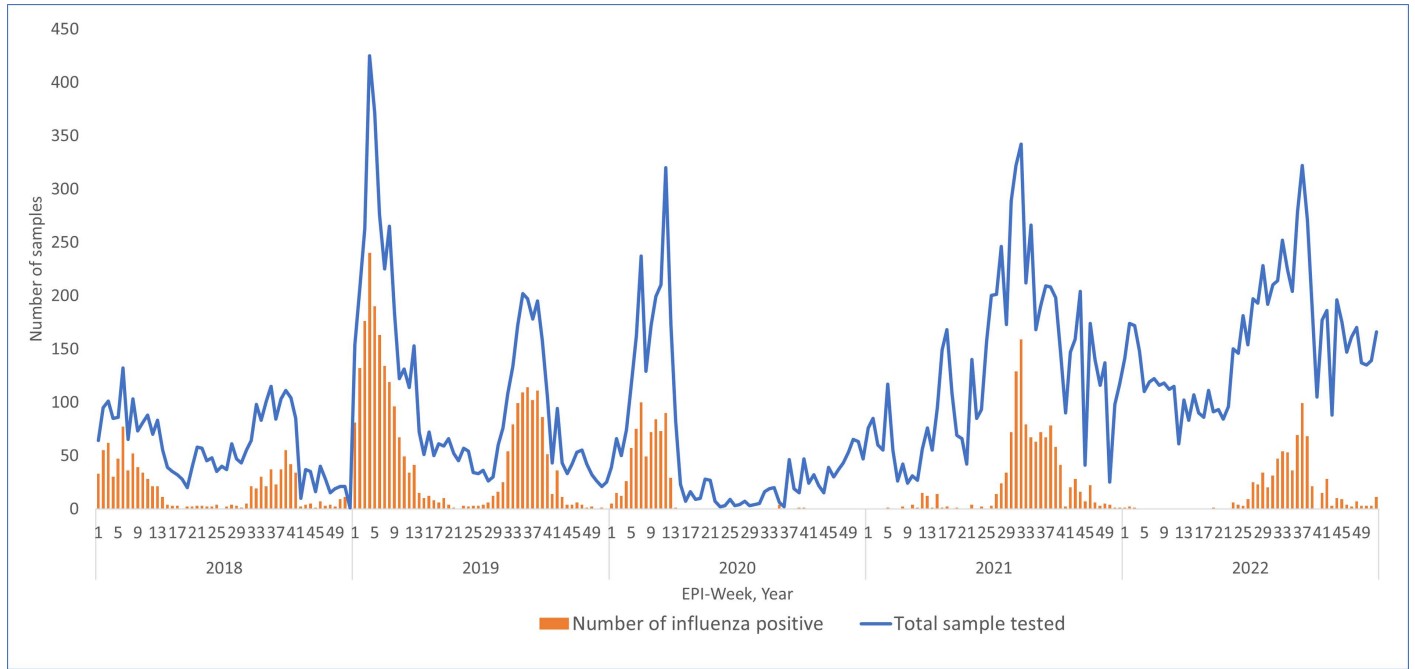

**Fig 3. Week wise total sample tested and number of positive influenza cases from year 2018 to 2022.**

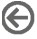

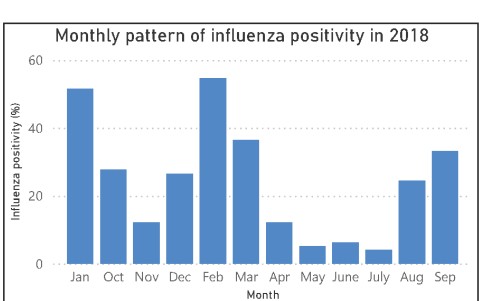

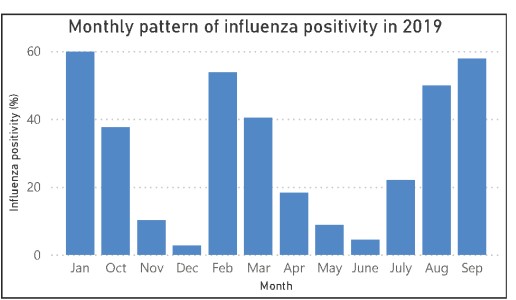

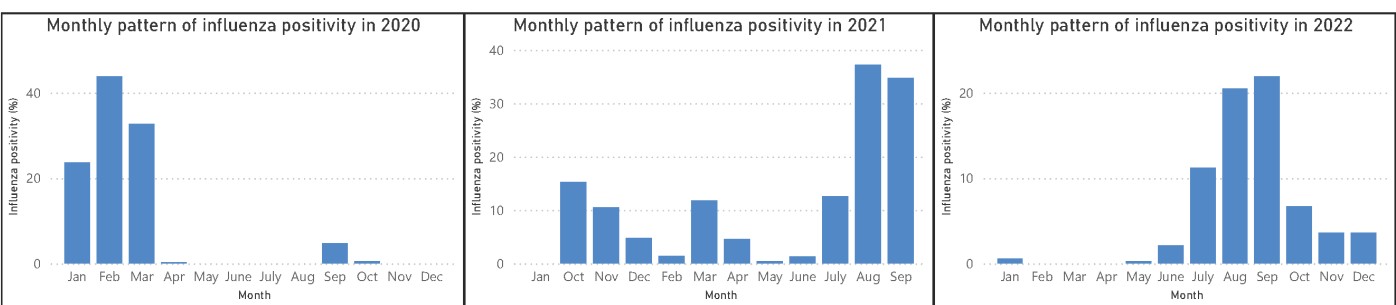

**Fig 4. Monthly pattern of influenza positivity from 2018 to 2022.**

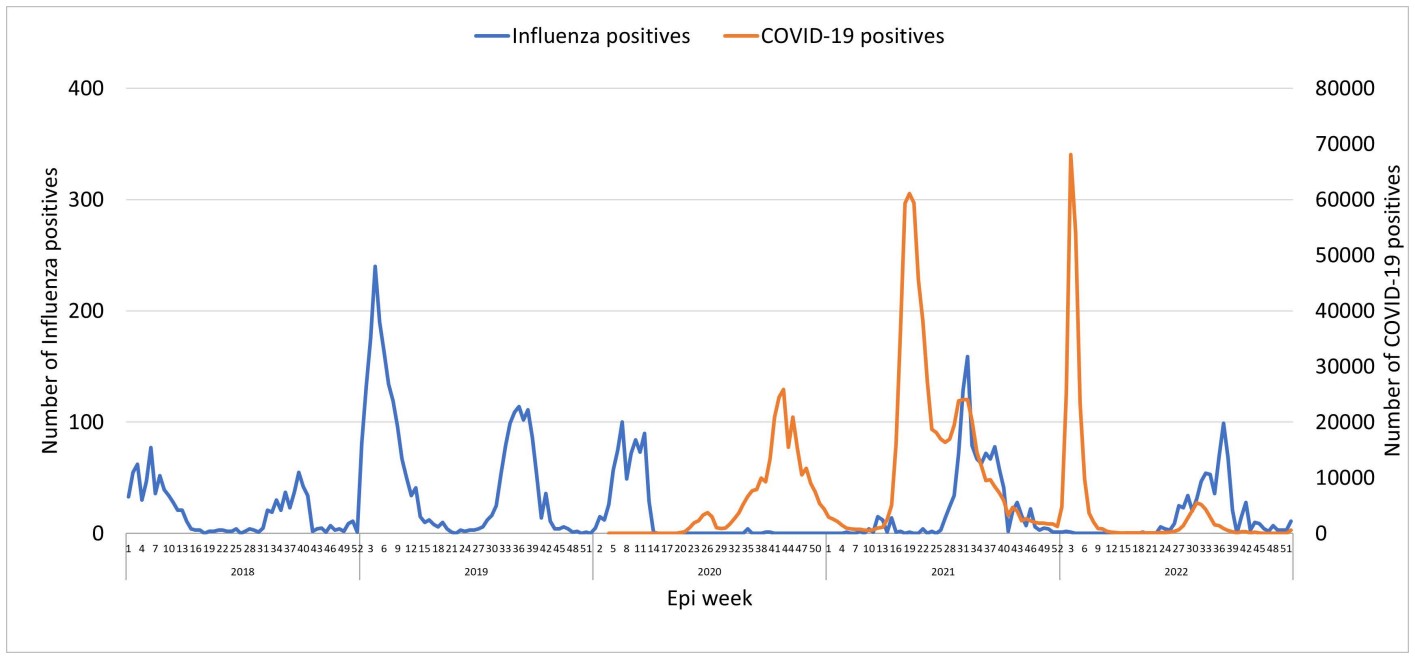

**Fig 5. Influenza and COVID-19 Positivity of Nepal from 2018 to 2022.**

**Table 3. Distribution of Influenza in different age group, gender, and clinical cases from NIC data (N = 18688).**

| | | Pre COVID-19 (2018–2019) | | | | During COVID-19 (2020–2022) | | | |
|---|---|---|---|---|---|---|---|---|---|
| | | Negative | Positive | Total | (%) Positivity | Negative | Positive | Total | (%) Positivity |
| Age group in year | <5 | 530 | 99 | 629 | 15.7 | 1117 | 129 | 1246 | 10.4 |
| | 5–14 | 444 | 155 | 599 | 25.9 | 1610 | 155 | 1765 | 8.8 |
| | 15–64 | 2368 | 1248 | 3616 | 34.5 | 7873 | 677 | 8550 | 7.9 |
| | ≥65 | 836 | 284 | 1120 | 25.4 | 1095 | 68 | 1163 | 5.8 |
| Case | ILI | 1109 | 711 | 1820 | 39.1 | 6066 | 456 | 6522 | 7 |
| | SARI | 3069 | 1075 | 4144 | 25.9 | 5629 | 573 | 6202 | 9.2 |
| Gender | Female | 1942 | 865 | 2807 | 30.8 | 4750 | 449 | 5199 | 8.6 |
| | Male | 2236 | 921 | 3157 | 29.2 | 6945 | 580 | 7525 | 7.7 |

during the pandemic years, the distribution of ILI and SARI cases was almost the same, with SARI cases being slightly higher. Gender wise, although male population was found to be more affected, no statistically significant differences were detected.

## Discussion

Our study has shown that during pre-COVID-19 years, influenza cases were seen throughout the year with dual peak, one in winter season (January to March) and another in rainy season (July to October). Other studies from Nepal in previous years have also shown year-round circulation of influenza virus with seasonal peaks [14,15]. The study by Wong NS et al from Hong Kong, China, also reported similar dual influenza peak: a winter peak that occur within initial 2 months of the year and a summer peak in the month of August or September [16].

In contrast, during the COVID-19 years a significant decline in influenza cases was observed in Nepal (p=<0.001). Correspondingly, global surveillance data have shown a decline in influenza cases in response to the COVID-19 pandemic [17–20]. In 2020, a smaller number of influenza tests were performed compared to previous year due to the prioritization of COVID-19 testing. At the beginning of the year 2020, 35% of influenza cases were reported between epi week 1 and 13, after which the number of tests gradually decreased. One of the main reasons for this decline was likely the nationwide lockdown announced by the Government of Nepal on March 24, 2020, with cessation of domestic travel, physical distancing, and use of personal protective measures [21]. Influenza testing significantly increased during 2021–2022, reflecting a renewed focus on influenza surveillance. Nepal adopted the WHO's recommendation in 2021 to use COVID-19 PCR-negative samples for influenza testing, helping to maintain influenza surveillance [12]. Additionally, in 2022, the national integrated sentinel surveillance for influenza and SARS-CoV-2 was expanded across provinces, which strengthened the surveillance system and increased the number of samples tested [22,23]. Despite this, the positivity rates remained notably low (16.7% in 2021 and 8.7% in 2022). A notable observation is that the rise in COVID-19 cases in Nepal corresponded with a decrease in influenza positivity during 2020–2022. A study by Young Lu et. Al in China also reported similar findings, showing a significant decline in influenza cases coinciding with the COVID-19 pandemic [24]. Similar trend was seen in various other countries like Japan, Thailand, USA, and Xinxiang China where seasonal influenza circulation was lower in 2020 than in preceding years [25–28]. Canada also reported an overall lower percentage of positive tests for all seasonal respiratory viruses, including influenza A and B, during 2020–2021 compared to pre-pandemic seasons [29]. Studies from East Asia and Southeast Asia have also observed a decrease in seasonal influenza cases following the emergence of COVID-19, possibly attributed to effective preventive measures such as mask-wearing, hand hygiene, and self-imposed physical distancing, which likely contributed to reducing influenza transmission [4,30].

In addition to a decreased positivity rate, COVID-19 has also affected the regular seasonal cycle of influenza circulation in Nepal. During the COVID-19 period, only a single peak of influenza was observed, in contrast to the regular biannual peak seen before the pandemic. Similar findings, with late and unusually prolonged influenza seasons have been reported in the Northen hemisphere [31]. Likewise, in Malaysia, a shift in influenza virus seasonality was observed, changing from a bi annual peak before the pandemic to no clear seasonal pattern during the pandemic, indicating a notable impact on the virus's seasonal dynamics [32].

The widespread use of public health interventions to prevent COVID -19 pandemic may be one of the factors explaining the rapid decline in influenza cases, as both the diseases share common route of transmission, which may significantly lower the risk of influenza exposure [19]. A study by Salina et al, on knowledge, attitude and practice on COVID-19 among general population in the eight most affected districts of Nepal showed that 93.3% of respondents were knowledgeable about overall COVID-19 preventive practices. However, these practices were reduced after the lockdown compared to during the lockdown [33]. Similar, studies in the USA have shown that implementing public health measures such as social distancing, travel restrictions, use of face masks, and frequent hand washing has contributed to these observations [34]. Astudy by Yiman Geng et al showed a significant reduction of the spread of the influenza virus following public health interventions for COVID-19 in China [35].

The COVID-19 pandemic posed a significant challenge to influenza surveillance, impacting the completeness, and geographical representativeness of data collection. During the early phase of the pandemic, sentinel influenza surveillance activities in Nepal declined substantially. To address this, SARS-CoV-2 PCR-negative samples were tested for influenza following WHO guidelines. However, the prioritization of respiratory samples for SARS-CoV-2 testing meant that only negative cases were screened for influenza, increasing the likelihood of missed dual-positive cases. These shifts in surveillance performance, driven by the pandemic response, represent a key limitation of this study.

Other limitation of our study is that we could only present limited epidemiological data, specifically age, gender and case-specific data available from the NIC, NPHL. Epidemiological data such as age and gender distribution are not available in RespiMart so complete data regarding these factors was not accessible. However, available data has shown that

there is a significant difference in the age -wise positive cases pre-COVID-19 and during COVID-19. Our results demonstrate higher positivity in the age groups of 15–64 during pre-COVID -19 and 0–14 during COVID-19.

In terms of Influenza B lineage determination, studies by Zoltan et al., Marios et al., and Paget et al have reported no new isolates of the Yamagata lineage of the influenza B virus since March 2020, suggesting it may no longer exist [36–38]. Interestingly, our study found that influenza B/Yamagata and influenza B/Victoria co-circulated until 2019, with influenza B/Yamagata predominating in 2018. However, from 2020 to 2022, subtype of influenza B subtypes was limited, and many cases were not further classified into these lineages. Despite this, available data clearly indicate a significant predominance of the Victoria lineage, with no reports of the Yamagata lineage. Due to the limited subtyping of influenza B cases, it is difficult to accurately ascertain whether the influenza B/Yamagata lineage is extinct.

## Conclusion

The COVID-19 pandemic and the PHSM implemented in response had a significant impact on the circulation of the Influenza virus in Nepal. As soon as the lockdown and other PHSM were enforced there was a significant decline in influenza circulation, altering the previously observed trend. Although the COVID-19 pandemic presented unprecedented challenges, it has also provided significant insights into the control of respiratory viruses. These lessons should be leveraged to enhance our approach to seasonal influenza and improve overall preparedness for future respiratory disease threats. Furthermore, we recommend that countries adapt their influenza surveillance systems during public health emergencies to ensure continued monitoring. Future research should focus on long-term impacts of the pandemic on influenza ecology, potential viral evolution under reduced circulation, and strategies to integrate lessons learned from COVID-19 into seasonal influenza management. Understanding how COVID-19 and influenza circulate together will be important for enhancing our preparedness for future pandemics and can help us develop better strategies for managing respiratory viruses simultaneously.

## Supporting information

**S1 File. FluMart Data from 2018 to 2022.**
(XLSX)

**S2 File. Raw data set for Table 3.**
(XLSX)

## Acknowledgment

We would like to express our sincere gratitude to WHO Country office Nepal for organizing manuscript writing workshop. This work represents the personal opinion of the authors and not that of the World Health Organization and the Ministry of Health and Population, Government of Nepal.

## Author contributions

**Conceptualization:** Smriti Shrestha, Priya Jha, Lilee Shrestha, Runa Jha, ArunKumar Govindakarnavar.

**Data curation:** Smriti Shrestha, Priya Jha, Rashmi Mulmi.

**Formal analysis:** Smriti Shrestha, Priya Jha, Rashmi Mulmi, ArunKumar Govindakarnavar.

**Investigation:** Smriti Shrestha, Priya Jha, Lok Bandhu Chaudhary.

**Methodology:** Smriti Shrestha, Priya Jha, Rashmi Mulmi, Lok Bandhu Chaudhary, Runa Jha, ArunKumar Govindakarnavar.

**Project administration:** Lilee Shrestha.

**Supervision:** Runa Jha, ArunKumar Govindakarnavar.

**Validation:** Smriti Shrestha, Priya Jha, Lilee Shrestha, ArunKumar Govindakarnavar.

**Writing – original draft:** Smriti Shrestha, Priya Jha.

**Writing – review & editing:** Smriti Shrestha, Priya Jha, Lilee Shrestha, Runa Jha, ArunKumar Govindakarnavar.

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
