## [Decision Letter · Decision Letter 0]

3 Jul 2024

PONE-D-24-03169Trend of Influenza before and during the COVID-19 pandemic in Nepal - a study from 2018 to 2022PLOS ONE

Dear Dr. Jha,

Thank you for submitting your manuscript to PLOS ONE. After careful consideration, we feel that it has merit but does not fully meet PLOS ONE’s publication criteria as it currently stands. Therefore, we invite you to submit a revised version of the manuscript that addresses the points raised during the review process.

We look forward to receiving your revised manuscript.

Kind regards,

Yoon-Seok Chung

Academic Editor

PLOS ONE

[No]. 

4. PLOS requires an ORCID iD for the corresponding author in Editorial Manager on papers submitted after December 6th, 2016. Please ensure that you have an ORCID iD and that it is validated in Editorial Manager. To do this, go to ‘Update my Information’ (in the upper left-hand corner of the main menu), and click on the Fetch/Validate link next to the ORCID field. This will take you to the ORCID site and allow you to create a new iD or authenticate a pre-existing iD in Editorial Manager. Please see the following video for instructions on linking an ORCID iD to your Editorial Manager account: https://www.youtube.com/watch?v=_xcclfuvtxQ.

5. Please amend the manuscript submission data (via Edit Submission) to include author Priya Jha2.

6. Please amend your authorship list in your manuscript file to include author Priya N/A Jha.

Additional Editor Comments (if provided):

Reviewers' comments:

Reviewer's Responses to Questions

**Comments to the Author**

1. Is the manuscript technically sound, and do the data support the conclusions?

Reviewer #1: Partly

Reviewer #2: Partly

Reviewer #3: Partly

Reviewer #4: No

2. Has the statistical analysis been performed appropriately and rigorously? 

Reviewer #1: Yes

Reviewer #2: No

Reviewer #3: Yes

Reviewer #4: No

3. Have the authors made all data underlying the findings in their manuscript fully available?

Reviewer #1: Yes

Reviewer #2: No

Reviewer #3: Yes

Reviewer #4: Yes

4. Is the manuscript presented in an intelligible fashion and written in standard English?

Reviewer #1: No

Reviewer #2: Yes

Reviewer #3: No

Reviewer #4: No

5. Review Comments to the Author

Reviewer #1: Thank you dear colleagues for this interesting research regarding changes in influenza circulation and relation with COVID-19 pandemic. Your findings are similar to numerous similar findings all over the world and bring a valuable piece, regarding Nepal in the global epidemiological puzzle.

Some important clarifications or nuancing of presented data are essential and minor editing interventions and typing error corrections are required.

It would be of more visual impact to split in two different figures the current Figure 1

A Figure 1 A with only Total cases tested and Positive cases [blue and magenta lines in present figure] in order to make more obvious the paucity of positive cases between week 17 in 2020 till week 5 in 2021. This change is needed in order to document also the shift of circulation from a bi-annual peak model in 2018 and 2019 to a single peak model in 2022.

It could also help to pinpoint on timeline of this graph when lock-down measures were implemented and reinforced and when these were stopped in order to generate potential epidemiological interference with air-borne viral diseases circulation.

You can comment in more detail interferences of with behavioural changes at populational level in relation with SARS-CoV-2 waves and non-pharmacological measures implemented and strength of their implementation.

A major proof that perception of danger is related to adherence to prevention measures is the upsurge of influenza positivity rates when local health authorities declared the end of COVID-19 related restrictions in our country in 2022 [Miron, V.D.; Bar, G.; Filimon, C.; Craiu, M. From COVID-19 to Influenza—Real-Life Clinical Practice in a Pediatric Hospital. Diagnostics 2022, 12, 1208. https://doi.org/10.3390/diagnostics12051208].

You should address in more detail the changes of influenza testing observed in 2021 and 2022 in your country and discuss why average number of total tests were almost double than before pandemic years studied outside of the presumed circulation period of influenza virus in your country. Because detection of shift in circulation model could be explained by significantly higher detections attributable to increased testing outside circulation season.

A separate Figure 1 B could help in better visualisation of vanishing positive B Yamagata results in 2020-2022 seasons, in concordance with available data. This approach is essential in order to improve discussions regarding potential implications in prevention strategies regarding influenza in a world free of Yamagata lineage [Caini S, Meijer A, Nunes MC, Henaff L, Zounon M, Boudewijns B, Del Riccio M, Paget J. Probable extinction of influenza B/Yamagata and its public health implications: a systematic literature review and assessment of global surveillance databases. Lancet Microbe. 2024 May 7:S2666-5247(24)00066-1. doi: 10.1016/S2666-5247(24)00066-1. Epub ahead of print. PMID: 38729197.]

Also it needs a more extensive approach to future changes in prevention strategy regarding immunization against influenza of various groups of people [Lin J, Li C, He W. Trends in influenza vaccine uptake before and during the COVID-19 pandemic in the USA. Public Health. 2023 Dec;225:291-298. doi: 10.1016/j.puhe.2023.10.028. Epub 2023 Nov 11. PMID: 37956641.]

Discussion section can be improved by inserting comments regarding behavioural changes generated by COVID-19 disease in relation with risk perception and SARS-CoV-2 circulation waves. Probably adherence to preventive measures were significantly different when tests and vaccines were not easy accessible (in 2020) as has been suggested by a recent 2024 paper [Keene K, Balasubramanian A, Potter A, Cioffredi LA. Maternal worry of children contracting COVID-19 predicts vaccine uptake in young children in Vermont. Vaccine X. 2024 Jan 19;16:100442. doi: 10.1016/j.jvacx.2024.100442. PMID: 38318233; PMCID: PMC10839441]. Do you have any data regarding mask usage, vaccine coverage, etc during pandemic in Nepal?

Please elaborate and insert relevant references to one of your major strengths of study presented in rows 181 - 182 "WHO recommendations and used COVID-19 PCR negative samples and retrospectively tested for Influenza to bridge the surveillance gap".

There are may data documenting relevance of age groups in influenza circulation during pandemic, most of them from Europe and USA []. Please insert a more detailed comment regarding the limitation regarding lack of extensive age related data in your paper

Minor clarifications and changes:

in rows 88-89 it is a repetition that has to be corrected "indicating a significant indicating a significant reduction (p value < 0.001) once the COVID-19 pandemic..."

in row 99 you are stating " During Pre-COVID-19 26% of influenza type A cases were" and in rows 102-103 you are stating "with Influenza type A predominating more than 95%" - please clarify and correct in row 99 where is another number (86% probably).

In row 110-111 you are stating that "only single peak was observed from July to October" during COVID-19 years. This is inaccurate because in 2020 no wave was documented [virtually no cases...] and in 2021 there were 2 waves with a small first wave during epidemiological weeks 5-15.

In References section you should improve citation style by inserting in references 1, 6, 8, 9 , 12 data when these sources were accessed and in reference 6 also site where this document is available.

Reviewer #2: Dear authors,

Thank you for submitting your manuscript. The paper provides an overview of influenza data for Nepal from 2018 to 2022. It is already well-established that influenza activity decreased during the COVID-19 pandemic, a trend observed in many countries. The article's content is quite general and does not offer new insights as of 2024. To enhance the paper, it would be valuable to explore the impact of the pandemic on other respiratory viruses, analyze severe acute respiratory infection (SARI) and influenza-like illness (ILI) cases in detail, and investigate potential influenza and COVID-19 coinfections. Additionally, incorporating data from 2023 and delving deeper into sentinel surveillance would strengthen the paper's contribution to the field. As it stands, the current state of the manuscript may not be suitable for publication.

Reviewer #3: The manuscript should review the scientific research elements and questions that need to be technically more precisely targeted with data that support the conclusions. The experiments must have been carried out rigorously, with controls, replication and a more appropriate methodology. Conclusions need to be drawn more accurately based on the data presented. Also this manuscript requires revision of the language.

Reviewer #4: Dear Authors,

I appreciate your effort to share the trends in the detection of the influenza virus before and during the COVID‐19 pandemic in Nepal.

However, you need to clearly state your objective, methods, and results. Based on the results, you should draw a conclusion and provide recommendations. There are fundamental errors in the development of the abstract and the manuscript (please follow the journal guideline).

The title of the manuscript does not represent the main story of the paper. Please consider rephrasing the title to reflect the content properly.

Please also follow standard methods to present your data. You may use the WHO average curve method to compare influenza virus seasonality/influenza positive trends before and during the pandemic periods. Consider using at least six years of data for the comparison (e.g., three years before the COVID‐19 pandemic and three years during the COVID‐19 pandemic).

There are several grammatical errors (punctuation, capitalization, and use of appropriate tense) in the manuscript. I believe your manuscript needs scientific editing to improve the scientific writing.

I hope to see an improved draft of your manuscript that meets the journal's standards.

Best wishes,

6. PLOS authors have the option to publish the peer review history of their article (what does this mean? ). If published, this will include your full peer review and any attached files.

**Do you want your identity to be public for this peer review?** For information about this choice, including consent withdrawal, please see our Privacy Policy .

Reviewer #1: **Yes: ** Mihai Craiu

Reviewer #2: No

Reviewer #3: No

Reviewer #4: No

---

## [Author Response · Author response to Decision Letter 0]

16 Sep 2024

Reviewers Comment and response

Reviewer 1:

Comment1: It would be of more visual impact to split in two different figures the current Figure 1

A Figure 1 A with only Total cases tested and Positive cases [blue and magenta lines in present figure] to make more obvious the paucity of positive cases between week 17 in 2020 till week 5 in 2021. This change is needed in order to document also the shift of circulation from a bi-annual peak model in 2018 and 2019 to a single peak model in 2022

Response: Fig 1 has been split into two figures Fig 1 A: Week wise total sample tested and number of influenza positive cases from year 2018 to 2022

Fig 1 B: Trend of Influenza during Pre-COVID-19 (2018-2019) and during COVID -19(2020-2022)

Comment 2 : You should address in more detail the changes of influenza testing observed in 2021 and 2022 in your country and discuss why average number of total tests were almost double than before pandemic years studied outside of the presumed circulation period of influenza virus in your country.

Response: Influenza testing significantly increased during 2021-2022, reflecting a renewed focus on influenza surveillance. Nepal adopted the WHO's recommendation in 2021 to use COVID-19 PCR-negative samples for influenza testing, helping to maintain the influenza surveillance. Additionally, in 2022, the national integrated sentinel surveillance for influenza and SARS-CoV-2 was expanded across provinces, which strengthened the surveillance system and increased the number of samples tested. This has been mentioned in the introduction section, line no 51 to 53, and further been added and elaborated in discussion section, line no 174-177.

Comment3: A separate Figure 1 B could help in better visualization of vanishing positive B Yamagata results in 2020-2022 seasons, in concordance with available data. This approach is essential in order to improve discussions regarding potential implications in prevention strategies regarding influenza in a world free of Yamagata lineage

Response: Figure 2 “Circulation of influenza subtype from 2018 to 2022” clearly illustrates the vanishing positive B Yamagata result in concordance with the available data. Also added in Fig 1B.

Comment 4: Discussion section can be improved by inserting comments regarding behavioral changes generated by COVID-19 disease in relation with risk perception and SARS-CoV-2 circulation waves. Do you have any data regarding mask usage, vaccine coverage, etc during pandemic in Nepal?

Response: Data from the available study related to behavioral changes was incorporated into the discussion section at line 196. However, there is limited information on mask usage, vaccine coverage, and similar factors during the pandemic in Nepal. Additionally, since the flu vaccine is not part of Nepal's routine immunization schedule, data on its coverage is scarcely available.

Comment 5: Please elaborate and insert relevant references to one of your major strengths of study presented in rows 181 - 182 "WHO recommendations and used COVID-19 PCR negative samples and retrospectively tested for Influenza to bridge the surveillance gap".

Response: This has been mentioned in the introduction section, line no 51 to 53, and further been added and elaborated in discussion section, line no 174-177.

Comment 6: Please insert a more detailed comment regarding the limitation regarding lack of extensive age-related data in your paper.

Response: The epidemiological data are not available from the collaborating center and RespiMart, thereby only data available from the NIC is analyzed and mentioned in methodology section line no: 81 to 84 and discussion line no 205 to 208. This age-related data doesn’t cover the entire country context age related influenza data hence the finding is not further elaborated and with other study. Kindly provide your feedback on same.

Comment 7: in rows 88-89 it is a repetition that has to be corrected "indicating a significant indicating a significant reduction (p value < 0.001) once the COVID-19 pandemic..."

Response: This has been corrected in result section line no 123 -124

Comment 8 in row 99 you are stating " During Pre-COVID-19 26% of influenza type A cases were" and in rows 102-103 you are stating "with Influenza type A predominating more than 95%" - please clarify and correct in row 99 where is another number (86% probably).

Response; Correction done in Result section line No; 120 to 122

Comment 9In row 110-111 you are stating that "only single peak was observed from July to October" during COVID-19 years. This is inaccurate because in 2020 no wave was documented [virtually no cases...] and in 2021 there were 2 waves with a small first wave during epidemiological weeks 5-15.

Response: Necessary correction done in result section line no: 136 to 137. This is also represented in Fig 1A

Response to Reviewer 2:

Comment 1: it would be valuable to explore the impact of the pandemic on other respiratory viruses

Response: There is unavailability of data for the other respiratory viruses besides SARS-CoV-2 and Influenza due to lack of testing for other respiratory viruses.

Comment 2 : analyze severe acute respiratory infection (SARI) and influenza-like illness (ILI) cases in detail.

Response: There is no significance difference s between SARI and ILI cases during Pandemic and the necessary details been mentioned in the result section line no146 to 149 and Table 2

Comment 3: investigate potential influenza and COVID-19 coinfections.

Response: No influenza and SARS-CoV-2 coinfection was detected during this study period.

Comment 4: Additionally, incorporating data from 2023 and delving deeper into sentinel surveillance would strengthen the paper's contribution to the field. As it stands, the current state of the manuscript may not be suitable for publication.

Response: As the study timeline mentioned from 2018 to 2022, thereby data of 2023 cannot be incorporated due to ethical clearance consideration.

Response to 3rd Reviewer:

Comment 1: Conclusions need to be drawn more accurately based on the data presented.

Response: Necessary changes have been made in the conclusion section as required

Comment 2: Also, this manuscript requires revision of the language.

Response: Necessary amendment has been done

Response to Reviewer 4: Clearly state objective, methods and result. Based on the results, you should draw a conclusion and provide recommendations. There fundamental errors in the development of the abstract and the manuscript.

Response: The revisions have been made in response to the recommendations provided.

Academic Comment and response

1. Seasonality (We cannot talk about the seasonality of influenza during pandemic years. This study must be ensured by excluding pandemic years because influenza activity is not normal during this period. We are talking here about) - line no 14

Response: As suggested, it is agreed that the term seasonality is not applicable. Pandemic years do not ascertain seasonality, so the entire sentence have been removed and information is rephased and restructure in the abstract.

2. Five years (in flu surveillance we generally we use the term "flu season" not "year" so there are four flu seasons)- line no 15

Response: Instead of writing five years, we have collectively mentioned the years "between 2018 and 2022"- line no 14

3. Materials and Methods (check font and punctuation you talked about the type and subtype of influenza viruses but you did not describe the method followed and the NIC protocol for the identification of influenza viruses)- line no 62

Response: Font and punctuation is checked. Method followed in NIC and its protocol is described and added in line no 89-100

4. descriptive cross-sectional (you said that your survey "is a retrospective study"; please check the type of your study) – line no 63

Response: The study type is checked, and is mentioned as retrospective cross-sectional study, line no 72

5. test (check please) – line no 65

Response: it is part of definition of RespiMart platform, this definition has been shifted in introduction section, line no 59-63 and necessary changes have been made.

6. system(Please apply capital letters)- line no 66

Response: The change has been made- line no 60

7. NIC, NPHL (don't start a sentence with an abbreviation) – line no 66

Response: The change has been made in all applicable sentences.

8. NIC (don't start a sentence with an abbreviation) – line no 67

Response: The change has been made in all applicable sentences.

9. sentinel and non- sentinel site (what is the type of influenza surveillance used sentinel or non-sentinel) – line no 68

Response: The type of influenza surveillance at NIC is sentinel surveillance. However, NIC also receive influenza data from other non- sentinel sites. Therefore, total data from both the sentinel and non-sentinel sites are uploaded in the RespiMart and incorporated in the study to cover overall picture of influenza circulation in Nepal during the study period.

10. Respimart (Please introduce Respimart in the main text) –line no 68

Response: Its introduction is added in introduction section -line no 59-63

11. Types and subtypes (A/H1, A/H3, Victoria, Yamagata) (Please respect the virus nomenclature (subtype; lineage, etc.)) – line no 71, 72)

Response: The change has been made - line no 74-76

12. Influenza (Please apply lowercase)- line no 73

Response: The necessary change is made.

13. Aggregate (aggregated) – line no 77

Response: The necessary change is made.

14. Influenza (Please apply lowercase) – line no 77

Response: The necessary change is made.

15. SARI and ILI (Please state the full name of the abbreviations SARI and ILI and add the case definitions) – line no 81

Response: The full name and definition is added in line no 82, 84-87

16. Age (remove the comma) – line no 86

Response: The comma has been removed - line no 104

17. Cases (remove the point) – line no 87

Response: point has been removed - line no 105

18. “Table 1 (remove the quotes) – line no 87

Response: removed

19. 2 (add table 2) – line no 87

Response: the information is maintained in table 2 so changes made accordingly, line no 105

20. Pre (Please apply lowercase) – line no 88

Response: It is applied wherever required

21. COVID (Please respect the nomenclature) – line no 88

Response: The correct nomenclature is added, line no 106

22. COVID (Please respect the nomenclature) – line no 88

Response: correct nomenclature is added, line no 106

23. indicating a significant (Please remove repetition) – line no 89

Response: Repetition is removed - line no 107

24. Pre (Please apply lowercase) – line no 91

Response: lowercase is applied wherever applicable

25. COVID (Please respect the nomenclature) – line no 91

Response: correct nomenclature given wherever required

26. COVID (Please respect the nomenclature)- line no 92

Response: correct nomenclature given wherever required

27. Negative (Please respect lowercase and uppercase) –Table 1

Response: It has been changed to required format - line no 119

28. H1, H3, H5 (respect the virus nomenclature and specify that it is the subtype) – Table 1

Response: correct nomenclature added, subtype and lineage is mentioned in table 1B, line no 122

29. NOT SUBTYPED (lowercase) – Table 1

Response: It has been written in correct format

30. VIC, YAM (respect the virus lineage nomenclature) – Table 1

Response: correct nomenclature is given

31. NOT SUBTYPED (lowercase and why not subtyped) – Table 1

Response: it is written in lowercase. Influenza subtype data of few influenza positive samples are not found as subtype has not been done for these samples due to reagent constraints, so to address these samples, they have been kept under subheading, “not subtyped”.

32. <0.001 (Please adjust table size and respect the same number of digits after the decimal point)

Response: the change has been made

33. PRE COVID (Please respect the nomenclature)- Table 1

Response: correct nomenclature is added.

34. DURING COVID (Please respect the nomenclature)- Table 1

Response: correct nomenclature is added

35. 7332(91.2) (please check your calculations and standardize the writing of the numbers in the table) – Table 1

Response: the change has been made

36. % (put in parentheses in front of the text)- table 2

Response: the change has been made

37. % (put in parentheses in front of the text)- table 2

Response: the change has been made

38. Age group (specify if age in year) – table 2

Response: the change has been made

39. 15 to 65 (I think it must be 64 and please check the calculations) – table 2

Response: it is 64, correction is done.

40. Influenza (the title is not relevant)- line no 97

Response: The title is changed to "distribution of influenza type and subtype", line no 128

41. Type (Please apply lowercase) – line no 97

Response: necessary changes has been made

42. The circulation of Influenza type A is found to be more prevalent in comparison to influenza type B throughout the given time period. During Pre-COVID-19 26% of influenza type A cases were detected, compared to 14% of type B cases, while during the COVID-19 period, there were 11.1% type A cases and 3.16% Type B cases(Please ensure language editing for your article)- line no 98

Response: the paragraph has been re-written in correct language format, line no 129-137

43. Pre- (remove the line) -line no 99

Response: necessary change has been made

44. In year 2020 and 2022 (Between 2020 and 2022)- line no 101

Response: This line has been removed, as the sentence show no relation with the result shown in the table.

45. has significantly dropped to less than 5% (shown in table 1), with Influenza type A predominating more than 95%(this is not consistent with the results of the table because your table is presented by subtype. Please place the table number towards the end of the sentence)- line no 102, 103

Response: The table 1 is split into table 1A and 1B. Table 1B clearly shows influenza type and subtype distribution, according to which, influenza B positivity is calculated in comparison to influenza A and result is mentioned in line no 134

46. Influenza A/H3(Please respecter la nomenclatiure influenza A H3N2)- line no 104

Response: correct nomenclature is given

47. Month-wise pattern of influenza positivity of year 2018 to 2022 (respect punctuation and check the language)- line no 107

Response: the necessary change has been made.

48. Distribution of influenza in age groups, gender, ILI and SARI cases (title is not relevant, it is a description of the population)- line no 112

Response: title has been changed.

49. adult age group 15-64 (it's not adulthood) – line no 113

Response: the term “adult” is removed and only the age group is mentioned.

50. pre COVID (check the nomenclature)- line no 114

Response: correct nomenclature is added

51. during COVID (2020-2022) (please check the title; add "and" and add the axis titles in the figure; put the figure title back under the figure towards the end of the article)- line no 118

Response: the necessary change is made

52. during (from)- line no 121

Response: the change is made

53. This has further added to the finding that influenza seasonality has variedin comparison to pre covid years.( studying influenza seasonality requires non-pandemic years to assert seasonality)- line no 151, 152

Response: This statement regarding seasonality has been removed as it is not applicable

Line no: 190.

54. One of the limitations of our study is that we were able to present limited epidemiological data, i.e, the age- and gender-specific data that was available in the NIC, NPHL databases only. Epidemiological data such as age and gender, distribution are not available in Flumart so complete data regarding the same was not available.( very long sentence to rephrase)- line no 162-165

Response: This sentence has been rephrased and presented according to our finding -line no 205-211

55. 15-64 (check the age groups in your table)- line no 167

Response: it has been checked, it is the correct age group

---

## [Decision Letter · Decision Letter 1]

27 Dec 2024

PONE-D-24-03169R1Trend of Influenza before and during the COVID-19 pandemic in Nepal - a study from 2018 to 2022PLOS ONE

Dear Dr. Jha,

Thank you for submitting your manuscript to PLOS ONE. After careful consideration, we feel that it has merit but does not fully meet PLOS ONE’s publication criteria as it currently stands. Therefore, we invite you to submit a revised version of the manuscript that addresses the points raised during the review process.

We look forward to receiving your revised manuscript.

Kind regards,

Victor Daniel Miron

Academic Editor

PLOS ONE

Journal Requirements:

Reviewers' comments:

Reviewer's Responses to Questions

**Comments to the Author**

1. If the authors have adequately addressed your comments raised in a previous round of review and you feel that this manuscript is now acceptable for publication, you may indicate that here to bypass the “Comments to the Author” section, enter your conflict of interest statement in the “Confidential to Editor” section, and submit your "Accept" recommendation.

Reviewer #4: All comments have been addressed

Reviewer #5: (No Response)

2. Is the manuscript technically sound, and do the data support the conclusions?

Reviewer #4: Yes

Reviewer #5: Partly

3. Has the statistical analysis been performed appropriately and rigorously? 

Reviewer #4: Yes

Reviewer #5: N/A

4. Have the authors made all data underlying the findings in their manuscript fully available?

Reviewer #4: Yes

Reviewer #5: No

5. Is the manuscript presented in an intelligible fashion and written in standard English?

Reviewer #4: No

Reviewer #5: Yes

6. Review Comments to the Author

Reviewer #4: Dear author,

Thank you for your thoughtful revisions and the resubmission of the manuscript. I appreciate your effort in addressing the initial review comments. The revised manuscript shows substantial improvements, with clearer arguments and improved formatting. However, a few aspects still require further clarification to ensure the manuscript meets its full potential. Below, I have outlined the points that need further attention:

Below, I have outlined the points that need further attention:

Abstract:

The objective could be redefined as:

“This study aims to understand the impact of public health and social measures implemented during the COVID-19 pandemic on influenza circulation in Nepal.”

Please provide data to support the following statement:

“Additionally, during the time of COVID-19 there has been a shift in the influenza bimodal seasonal pattern, with only one peak observed. Influenza type A consistently dominated, with variations in its subtype observed from year to year.”

While the conclusion statement is clear, it could be more impactful by linking it to broader implications for future public health measures. For example, focus on precise and actionable recommendations, such as how these findings inform public health policy or enhance pandemic preparedness.

Manuscript:

Introduction of the manuscript:

The objective statement could be refined as:

“This study aims to understand the impact of public health and social measures implemented during the COVID-19 pandemic on influenza circulation in Nepal from 2018 to 2022.”

The following statement should be moved to the Methods section:

“The period from 2018 to 2019 was considered the pre-COVID-19 period, while from 2020 to 2022 was considered the COVID-19 period.”

Methods:

Present case definitions for ILI and SARI in a separate paragraph under the subheading “Case Definition.”

In line 98, replace “A (H3N2/H1N1pdm09)” with “Influenza A(H1N1)pdm09 and Influenza A(H3N2)” for clarity.

Correct the sentence starting with “Subtyping of positive influenza B specimens...” for accuracy.

Mention the specific statistical tests used for analyzing trends and comment on their appropriateness for the dataset.

In lines 73–76, consider rewriting or splitting the sentence for better clarity, as it is currently hard to understand.

Notes:

Revise for grammatical consistency and clarity. For example, in lines 105–106, revise:

“The average positivity rate of influenza in pre-COVID-19 is 39% whereas during COVID-19 is 14%.” to “The average influenza positivity rate was 39% pre-COVID-19, compared to 14% during COVID-19.”

Avoid starting sentences with abbreviations. For example, in line 176, instead of:

“PHSM were enforced there was a drastic decline in Influenza circulation...”

revise to: “After public health and social measures (PHSM) were enforced, there was a drastic decline in influenza circulation...”

Avoid writing “influenza” with a capital letter unless it is the first word of a sentence.

Ensure consistent use of proper punctuation and capitalization throughout the manuscript (e.g., “During COVID-19” instead of “DURING COVID-19”).

By incorporating these suggestions, the manuscript will achieve greater clarity, coherence, and adherence to scholarly standards.

Reviewer #5: The author’s conclusion largely relies on surveillance data, which was omitted from the text. Therefore, I would recommend that the author provide more information about the surveillance system, including the number of sites, locations, etc in the Methods section.

The changes in the performance of the surveillance system may affect the author’s conclusion. Therefore, I would strongly recommend the performance changes of surveillance during the COVID-19 pandemic as a limitation of the study, referencing two literatures (PMID: 39173559 and PMID: 39045828).

Please elaborate on other studies conducted in Asian countries in the discussion.

7. PLOS authors have the option to publish the peer review history of their article (what does this mean? ). If published, this will include your full peer review and any attached files.

**Do you want your identity to be public for this peer review?** For information about this choice, including consent withdrawal, please see our Privacy Policy .

Reviewer #4: No

Reviewer #5: No

---

## [Author Response · Author response to Decision Letter 1]

9 Feb 2025

Please find the comments and response from the reviewer as follows:

Reviewers Comment and response

Reviewer #4

1. Comment : Abstract:

The objective could be redefined as: “This study aims to understand the impact of public health and social measures implemented during the COVID-19 pandemic on influenza circulation in Nepal.”

 Response: Added, line no: 12-13

2. Comment: Please provide data to support the following statement: “Additionally, during the time of COVID-19 there has been a shift in the influenza bimodal seasonal pattern, with only one peak observed. Influenza type A consistently dominated, with variations in its subtype observed from year to year.”

 Response: The data supporting this is provided in the Fig 1A and Fig 1B

3. Comment: While the conclusion statement is clear, it could be more impactful by linking it to broader implications for future public health measures. For example, focus on precise and actionable recommendations, such as how these findings inform public health policy or enhance pandemic preparedness.

 Response: Added, line no: 27-30

4. Comment: Manuscript:

Introduction of the manuscript: The objective statement could be refined as:“This study aims to understand the impact of public health and social measures implemented during the COVID-19 pandemic on influenza circulation in Nepal from 2018 to 2022.”

 Response: Added, line no 71-73

5. Comment: The following statement should be moved to the Methods section:“The period from 2018 to 2019 was considered the pre-COVID-19 period, while from 2020 to 2022 was considered the COVID-19 period.”

 Response: Added, line no 76-77

6. Comment: Method:

Present case definitions for ILI and SARI in a separate paragraph under the subheading “Case Definition.”

 Response: separate paragraph for case definition given. Line no: 92-97

7. Comment: In line 98, replace “A (H3N2/H1N1pdm09)” with “Influenza A(H1N1)pdm09 and Influenza A(H3N2)” for clarity.

 Response: It is PCR reagent name provided by CDC for molecular detection of influenza Hence, the name is kept as it is (CDC Influenza Virus Real-Time RT-PCR panel, Influenza A (H3N2/H1N1pdm09) subtyping panel).

8. Comment: Correct the sentence starting with “Subtyping of positive influenza B specimens...” for accuracy.

 Response: necessary changes made. Line no: 107

9. Comment: Mention the specific statistical tests used for analyzing trends and comment on their appropriateness for the dataset.

 Response: The specific statistical tests used for analyzing trends was Chi-square test to test the association between two categorical variables (i.e influenza positive, negative and pre and during COVID-19 trends mentioned in line number 81 to 88.

10. Comment: In lines 73–76, consider rewriting or splitting the sentence for better clarity, as it is currently hard to understand.

 Response: Necessary changes made. Line no: 81-86

11. Comment: Revise for grammatical consistency and clarity. For example, in lines 105–106,revise:“The average positivity rate of influenza in pre-COVID-19 is 39% whereas during COVID-19 is 14%.” to “The average influenza positivity rate was 39% pre-COVID-19, compared to 14% during COVID-19.”

 Response: Necessary revision done and changes made. Line no: 115-116

12. Comment: Avoid starting sentences with abbreviations. For example, in line 176, instead of: “PHSM were enforced there was a drastic decline in Influenza circulation...” revise to: “After public health and social measures (PHSM) were enforced, there was a drastic decline in influenza circulation...”

 Response: revise to: Necessary changes made. Line no: 211-212

13. Comment: Avoid writing “influenza” with a capital letter unless it is the first word of a sentence.

 Response: Changes made where applicable.

14. Comment: Ensure consistent use of proper punctuation and capitalization throughout the manuscript (e.g., “During COVID-19” instead of “DURING COVID-19”).

 Response: Necessary changes made.

Reviewer #5:

1. Comment: The author’s conclusion largely relies on surveillance data, which was omitted from the text. Therefore, I would recommend that the author provide more information about the surveillance system, including the number of sites, locations, etc in the Methods section.

 Response: More information provided on surveillance system in method section. Line no: 78-81,

Introduction section. Line no: 57-61

Discussion section: 185-191

2. Comment: The changes in the performance of the surveillance system may affect the author’s conclusion. Therefore, I would strongly recommend the performance changes of surveillance during the COVID-19 pandemic as a limitation of the study, referencing two literatures (PMID: 39173559 and PMID: 39045828).

 Response: Addressed as the limitation of the study in the discussion section in line number 219 to 226

3. Comment: Please elaborate on other studies conducted in Asian countries in the discussion.

 Response: Studies conducted in east and southeast Asia, Malaysia reported in line no: 199-202 and line no: 204-206.

We have completed the revisions and made the necessary adjustments to address the reviewers' and academic concerns, aiming to improve the manuscript. We trust the revised version meets with your approval. Please let us know if any further clarification is needed. Thank you once again for your consideration.

---

## [Editor Report · Decision Letter 2]

17 Mar 2025

Trend of Influenza before and during the COVID-19 pandemic in Nepal - a study from 2018 to 2022

PONE-D-24-03169R2

Dear Dr. Jha,

We’re pleased to inform you that your manuscript has been judged scientifically suitable for publication and will be formally accepted for publication once it meets all outstanding technical requirements.

Kind regards,

Victor Daniel Miron

Academic Editor

PLOS ONE

---

## [Editor Report · Acceptance letter]

PONE-D-24-03169R2

PLOS ONE

Dear Dr. Jha,

I'm pleased to inform you that your manuscript has been deemed suitable for publication in PLOS ONE. Congratulations! Your manuscript is now being handed over to our production team.

Kind regards,

on behalf of

Dr. Victor Daniel Miron

Academic Editor

PLOS ONE